# The Domestic Tyranny of Haunted Houses in Mary Wilkins Freeman and Shirley Jackson

**Christine Junker**

English Department, Wright State University—Lake Campus, Celina, OH 45822, USA;
christine.wilson@wright.edu

**Abstract:** Mary Wilkins Freeman and Shirley Jackson, though writing in different time periods, are both invested in recuperating domesticity and using their work to imagine what domesticity removed from the context of marriage and children can offer single women. Both authors assert that emplacement within domestic enclosure is essential to securing feminine subjectivity, but their haunted house narratives undermine that very emplacement. Freeman's stories, "The Southwest Chamber" and "The Hall Bedroom" anticipate Jackson's more well-known *The Haunting of Hill House* in the way that unruly domesticity threatens the female character's emplacement. Their haunted house narratives show that neither Freeman nor Jackson, for all that they are subversive in some ways, wants to dissolve the traditional ideological constructs of domesticity; instead, they want these ideologies to work in the culturally promised patriarchal fashion. Reading their haunted house narratives together reveals the dynamics and tensions of a domesticity that is fluid, entangled, and vibrant and the feminist potential such sites engender, even if the characters and texts in question cannot fully realize that potential.

**Keywords:** Shirley Jackson; Mary Wilkins Freeman; Feminism; Haunted Houses; The Haunting of Hill House; domesticity in literature

---

## 1. Introduction

Writing nearly sixty years apart and in dramatically different cultural landscapes, Mary Wilkins Freeman and Shirley Jackson share the conviction that it is essential for women to be able to shape their own spaces, just as they are shaped by those spaces. Both use their works to explore the tyranny—and promise—of the home and domesticity. They are linked by their hope that a different kind of domesticity can engender a different kind of feminine subjectivity. Both, too, are drawn to the idea that embracing "blessed singlehood" will lead to blissful domesticity but are simultaneously anxious that this ideal is inaccessible. Appropriately enough, within their haunted house narratives we can see what they imagine to be the perils for independent women, which revolve primarily around the loss of home and domestic enclosure and the resulting fragmentation of a sense of self and identity.

Freeman, best known as a regionalist writer in the nineteenth and earliest twentieth century, anticipates Jackson's concerns regarding gender and domesticity. In Freeman's haunted house short stories "The Southwest Chamber" and "The Hall Room," economic and material concerns dominate the landscape, and the haunting of domestic spaces renders the single women in her texts homeless and ungrounded. Jackson's *The Haunting of Hill House* (1959), despite the progress made by First Wave feminism and then-nascent Second Wave feminism, demonstrates that though material conditions may have changed, independent domesticity is no more available to her protagonist than it was for Freeman's. For both writers, the figure of the single woman with a room of her own remains haunted and out of reach.

Reading the opening of Jackson's *The Haunting of Hill House* against Freeman's most frequently anthologized story, "A New England Nun" (1887) articulates the connection between domestic space and feminine subjectivity that shapes the central concerns in Jackson and Freeman's works on haunted domesticity. On her way to Hill House, Eleanor passes a house with pillars and "a pair of stone lions" and thinks that "perhaps she might live there, dusting the lions each morning and patting their heads good night." Eleanor imagines that she has lived in that house, served by an older woman, eating her dinner "alone" and sleeping under a "canopy of white organdy" (Jackson 1984, p. 9). A few miles down the road, when she passes oleanders, Eleanor again enters the land of fantasy where she might slip between "magical gateposts" to find a fairytale garden and cottage. Eleanor returns to these domestic fantasies throughout the novel, even going so far as to tell Theodora that she lives in an apartment with "little stone lions." She explains, "Everything has to be exactly the way I want it, because there's only me to use it . . . " (Jackson 1984, p. 88). The emphasis for Eleanor is on living alone; she does not fantasize about a domesticity that includes marriage or children, or even other family members (Roberts 2017, p. 79).

Louisa Ellis, in Freeman's "A New England Nun," shares Eleanor's desire to live alone, with everything placed "exactly the way [she] want[s] it" (Freeman 1991, p. 109). What Eleanor creates in fantasy, Louisa constructs in reality. From her domestic objects like sewing needles, scissors, and multiple aprons, all of which have become "a very part of her personality" (Freeman 1991, p. 109), to her carefully arranged table—she always uses her china since she is a "guest to her own self" (Freeman 1991, p. 110)—Louisa arranges her space, and by extension her life and identity, to match her every desire. The threat to Louisa's domestic bliss is marriage, and the story focuses on Louisa's aversion of that crisis, ending with Louisa, "all alone by herself that night, we[eping] a little, she hardly knew why; but the next morning, on waking, she felt like a queen who, after fearing lest her domain be wrested away from her, sees it firmly insured in her possession" (Freeman 1991, p. 124). Like Eleanor, for Louisa, feminine autonomy and fulfillment are possible only if she has her own domestic space (Daniel 1996, p. 71; Glasser 1996, p. 35).

The opening of *The Haunting of Hill House* and "A New England Nun" show just how much both Jackson and Freeman were invested in a particular kind of idealized domesticity. Both imagine that if domesticity can just be disentangled from marriage and motherhood, it can provide a refuge from patriarchy and the concerns of the market. The danger of domesticity in these texts is that it is so enticing and desirable that one might give up the very thing that the home is meant to provide: freedom. What is reactionary about both Freeman and Jackson, rather than progressive and liberating, is that both seem to make a claim that feeling "at home" is rewarding enough to be worth any price, including the female sense of identity and self. Both imagine that a protected domestic space is what is necessary for feminine autonomy, but this fantasy is in direct tension with the anxiety that a protected sense of space is in direct conflict with feminine autonomy. These texts construct this tension as a uniquely feminine problem, since the male characters demonstrate no such concern or quandary. Finally, both authors, despite the vast historical "progress" made between the time periods in which their texts are written, express the anxiety that it is not heterosexual marriage or the presence of a patriarch that makes space un-inhabitable for women, but rather that the very notions and desire for domestic space that is un-inhabitable. Read together, both author's haunted house texts make a compelling critique of the fundamental idea of "home" and "domesticity," albeit unintentionally, because their haunted house narratives express the nihilism of the desire to be housed at all costs, especially for women.

Jackson and Freeman try to recuperate domesticity by imagining domestic space as a way to preserve feminine autonomy. Freeman imagines Louisa as a character who has the agency to choose herself and choose her space, and thus can be safe from the patriarchal norms of her culture. Moreover, Freeman embraces the notion that domesticity is not only a separate but an alternative space to the market. Domesticity is enacted ideally—it creates an impermeable boundary that protects its inhabitants from concerns of the "world," including the market and patriarchy, entirely. Jackson, writing fifty years later, imagines a similar domestic space that will protect Eleanor from the threatening

forces "outside." Yet, both authors' argument that domesticity can be a subversive space is hardly straightforward or uncontested.

As a literal setting for many works written by women during the time period, as well as an ideological construct that is viewed as simultaneously fulfilling and progressive and also oppressive, conservative, and colonizing, domesticity occupies a vexed territory within nineteenth- and twentieth-century American literature. Although Freeman is not writing at the height of the cult of domesticity, her work is nonetheless influenced by the ideology that conflates women's identities with their space, so that the performance of domesticity becomes intertwined with gender identity. Similarly, Jackson is writing during a resurgence of post-war domesticity in America (Hague 2005, p. 83). Freeman and Jackson imagine they can sever the connection between domesticity and (heterosexual) marriage and motherhood, while preserving the components of domesticity they find more amenable to their objectives: environmental determinism and the doctrine of separate spheres.

Freeman and Jackson are writing in the broader contexts of national and patriarchal ideologies. Klimasmith (2005) argues that "The nation itself had been constructed in part on foundations of architectural, or at least environmental determinism; the idea that built space could shape its inhabitants has marked the American landscape and shaped our notions of citizenship from the seventeenth century on" (p. 2). The belief that we are shaped by our environments grants home-spaces significance and endows the cult of domesticity with a peculiar kind of power; by prescribing gendered spaces and gendered performances within those spaces, identity becomes spatialized, and the everyday maintenance of a household—sweeping the floor, dusting knickknacks—becomes imbued with importance beyond simply maintaining tidiness. Performing domesticity becomes inextricable from performing femininity. Moreover, this performance suggests that feminine subjectivity is intertwined with women's spaces, which the culture views as domestic. For Freeman and Jackson, the idea of environmental determinism emerges through their insistence that domesticity should be emplaced. Casey (2009) uses the term "emplacement" to describe the way subjects and places are mutually constitutive and describes emplacement as "an ongoing cultural process" (p. 31). Subjects shape their dwelling places as much as they are shaped by them, and emplacement is always about "being concretely placed" and is "intrinsically particular" (Casey 2009, pp. 23, 120) Characters in Freeman's and Jackson's fiction affirm Casey's assertion that emplacement is what assures subjects that they "are embodied-in-place" and will not have to "suffer from the *horror vacui* that afflicted . . . place-deprived souls in the early Modern era" (italics original, Casey 2009, p. 104). In my reading of Freeman and Jackson, being emplaced within very specific domestic spaces is essential to women's own development as subjects, just as their own subjectivity is essential to their creation of these (fantasy) spaces.

Closely related to environmental determinism is the doctrine of separate spheres, which emerged in the United States in the late eighteenth century.[1] Separate spheres doctrine, which was preached in churches, reinforced in advertising, literature, and pamphlets, prescribed that it is "natural" that men and women occupy two separate spheres: the marketplace and public spaces on the one side, domestic and home spaces on the other. While arguably more of an ideal than a reality and, even then, an ideal limited to only privileged members of society, the influence of separate spheres worked concurrently with the cult of domesticity to spatialize identity. The doctrine of separate spheres and its implications for women plays a prominent role within the tradition of the Female Gothic. Kate Ferguson Ellis identifies the eighteenth-century Gothic novel as a site of critique of the doctrine of separate spheres because these works illuminate the tension between the home as a "refuge from evil" and the home as a "prison" (Ellis 1989, pp. xiii, xv). Carpenter and Kolmar (1991) argue that the figure of the haunted house permeates the writing of women writers in this genre because the doctrine of separate spheres so closely identified women with domestic spaces, and women writing in the

---

[1]	This doctrine was not limited to the United States, but my focus is on how this particular set of ideologies emerged in an American context as an aspect of national discourses surrounding environmental determinism.

supernatural genre "inherited a series of themes and images—of women victimized in their own homes, of women dispossessed of home and property, of the necessity of understanding female history, and of the bonds between women, living and dead, which help to ensure women's survival" (pp. 10, 14). Likewise, Bailey (1999) calls the (haunted) house a "potent symbol" that's a "primary marker of class and our central symbol of domesticity" (p. 8). While Weinstock (2008) acknowledges the importance of uncanny domestic space in American women's supernatural writing in the nineteenth century, he also asserts that two of the primary themes that haunt this writing are "marriage and motherhood" (p. 22). Indeed, within the Female Gothic, haunted domestic space is often inextricable from an accompanying marriage and the duties of mothering, and critiques of domesticity are partially, at least, critiques of the way marriage and motherhood within a patriarchal society place profound limitations on women.

Freeman and Jackson depart from this tradition by focusing their narratives on what haunts the spaces of spinsters, women who have potentially escaped some of the most restrictive aspects of patriarchy. These two women writers recognize the limitations of environmental determinism and the separate spheres doctrine, but they also try to harness tendrils of subversive potential within these concepts.[2] Both authors assert that emplacement within domestic enclosure is essential to securing feminine subjectivity, but their haunted house narratives undermine that very emplacement. Freeman's stories chart out the material barriers to home for single women, while Jackson's novel suggests that the barriers to independent identity and domesticity are primarily interior and psychological. For both of these authors, in other words, a blissful and independent domesticity is deeply desired, but their haunted house works demonstrate an anxiety that the promises of domesticity are false and that escaping heterosexual unions does not mean that one can escape the oppressiveness of patriarchy. Reading their haunted house narratives together reveals the dynamics and tensions of this domesticity and the feminist potential such sites engender, even if the characters in question cannot fully realize that potential.[3]

## 2. Strangling Bedcaps: Mary Wilkins Freeman's Haunted Domesticity

Over the course of her decades long writing career, Mary Wilkins Freeman's stories are preoccupied, obsessed even, with houses whose modesty belie the layers of meaning behind these unassuming spaces that are often tense with contradiction. Not surprisingly, then, Freeman's work is well-known for its depictions of domesticity, and most feminist critics read her version of domesticity as paying homage to the expectations of sentimentalism during the time period in which she was writing, while also providing her female characters with a sense of liberation and autonomy, in spirit if not in fact (Mann 1998, p. 42). In keeping with Brown's (1990) depiction of domestic individualism, wherein the self is "nearly synonymous with domesticity" (p. 3), Freeman's protagonists find a sense of a self, only insofar as they have a domestic space within which to house that self. In a time period when literature by and about women is dominated with themes of marriage and children within a

---

[2]   Feminist interpretations of domesticity, whether in the nineteenth or twentieth centuries have varied widely. Some critics, like Amy Kaplan, in "Manifest Domesticity," argue that domesticity was integral to the larger national project of colonizing the "foreign," revealing how domesticity was never just about what happened "in the home" (*American* Literature 70.3 1998, 563). Jane Tompkin's argument that the home acts not as "a retreat or refuge" from the outside world, but rather an "economic *alternative* to that world" and, moreover, an alternative that subverts the dominant values of American society offers one of the most compelling cases for the redemptive power of domesticity and its feminist potential ("Sentimental Power: *Uncle Tom's Cabin* and the Politics of Literary History" *Uncle Tom's Cabin*, edited by Elizabeth Ammons (New York City: W. W. Norton & Company 1994). Other scholars, such as Laura Wexler, argue that domesticity and sentimentalism extend and reinforce the project of cultural imperialism (*Tender Violence: Domestic Vision in an Age of U.S. Imperialism* (Chapel Hill: University of North Carolina Press, 2000). However domesticity is interpreted, as Lora Romero points out, it is a lens through which to view the complicated structures of gender and power in nineteenth and twentieth century America and domesticity may be used in ways that are both, sometimes at the same time, progressive and reactionary (*Home Fronts: Domesticity and its Critics in the Antebellum United States* (Durham: Duke University Press, 1997).
[3]   See Avril Horner and Sue Zlosnik, "Female Gothic" in *Teaching the Gothic*, edited by Anna Powell and Andrew Smith (New York: Palgrave Macmillan 2006) 107. Also see Lynette Carpenter and Wendy K. Kolmar in *Haunting the House of Fiction: Feminist Perspectives on Ghost Stories by American Women* (Knoxville: University of Tennessee Press, 1992).

single-family home, Freeman creates fictional spaces in which women are independent and often live alone or in the nurturing company of other women.

Though representative of a brief period and body of works for Freeman, her supernatural tales offer a forceful critique of the pinnacle of nineteenth-century womanhood that so dominated her earlier work: domesticity. This is in keeping with Weinstock's (2008) argument about the often-unacknowledged American Female Gothic tradition; Weinstock argues that supernatural fiction by women, written primarily between 1850 and 1930, allows women who had traditionally written sentimental, domestic works to critique the patriarchal political ideologies of the time period (p. 19). In her supernatural stories, the domestic sphere has become all too permeable, in both realistic (economic concerns, for example) and supernatural ways (namely, ghosts and strangling bedcaps). Freeman's haunted house stories provide a sharp contrast to the idealism of her earlier work, especially regarding the spatial and personal possibilities of domesticity and independence for women. These stories suggest that Freeman's faith in the redemptive powers of domesticity had taken a serious blow by 1903 and that her notion of the potentialities of feminine independence had become haunted with the realization that domestic space, and hence female agency, was not impervious to the constraints of the economic realities of patriarchy. Turning to the supernatural allowed Freeman "to give oblique expression to disturbing personal issues without fear of exposure" (Fisken 1991, p. 60) and to do what she could not do in any other way: express the deep fear that the place, literal and figurative, outside of patriarchy that her earlier works found for her female characters was untenable and that there is no escape, or even refuge, from the dictates of heterosexuality and marriage.

The protagonists of "The Southwest Chamber" and "The Hall Bedroom" (published originally in 1903) are single women who run boardinghouses. In the parlance of the late nineteenth century, this economized domesticity represents the encroachment of the public into the private sphere and effectively codes the women as "homeless" (Gamber 2002, p. 184). In the stories "The Southwest Chamber" and "The Hall Bedroom," the stakes of maintaining boarders are high: without their economic contributions, the protagonists will find themselves literally and metaphorically homeless. The presence of spinsters is common in Freeman's work, yet these stories diverge because instead of being able to successfully make a habitable space for themselves, these characters cannot, primarily because domestic items betray them, and economic concerns refuse to remain where they belong within the doctrine of separate spheres (which is to say, not in the house at all). In both stories, spinsters and widows are left to fend for themselves in an economic and cultural world that does not have space for single women.

From the beginning, these stories are about domestic spaces whose borders are all too permeable, and the hauntings demonstrate the pollution of these spaces by showing the horror that occurs when the outside world enters the domestic, causing even the most commonplace facets of domesticity become wild and uncontrollable. To twenty-first-century readers, however, the manifestations of hauntings in "The Southwest Chamber" seem almost comical because they are so mundane. Dresses move in and out of the Southwest room, sometimes appearing in closets or strewn over chairs, sometimes with their arms mysteriously sewn together; brooches transform from the familiar pearls to entwined locks of hair; the bedspread vacillates between a pattern of roses and a flock of peacocks. Water pitchers empty themselves, and a malicious sleeping cap ties itself to the head of an innocent woman slumbering in the chamber, magically reassembling itself even after it has been cut to pieces. Each instance of haunting reminds the reader that domestic, feminine items cannot be trusted. Whether it is ordinary (feminine) apparel, sewing baskets, or decor, nothing in this room is stable, and regular domestic life has been corrupted in a way that is meant to terrify the women in the story. The realm of the domestic has been infiltrated not just by the market, but by the natural world—objects that should be inert have become animated, uncanny. This animated domesticity space leads to expulsion and displacement, particularly for women.

For instance, when Sophia warns Amanda that, "If folks are going to act like fools we shall never be able to keep this house," it makes the stakes of controlling the domestic clear: if they cannot

subdue the uncontrollable elements, it threatens the Gill sisters' ability to provide boarding and, in consequence, to keep their house at all (Freeman 1986, p. 142). The hauntings thus call attention to the ways in which running a boardinghouse is already a fraught prospect. As Gamber (2002) points out, the daily activities that took place in boardinghouses "furnished glaring examples of corrupted domesticity," an apt description for the way domesticity is enacted in the Gill household (pp. 180–81). It is corrupted because rather than carrying out their domestic roles as an act of love and devotion to their family, the Gill sisters enact domesticity to make a profit, and any domestic failures impede their chance at success in that endeavor. If capitalism is a corrupting force, so too is the supernatural. Within the story, their failure to perform domesticity effectively threatens not just their sense of emplacement, but more importantly within the story, their ability to be emplaced at all. The Gill sisters are not represented as inadequate housekeepers, but rather as women who are being undermined, perhaps punished, by the very domestic objects that they are trying so hard to preserve.

The case of the one lone male character, Reverend Dunn (whose very name associates him with the economic world of bill collections), exemplifies how gender dictates the experience of the haunted domesticity of the Southwest chamber. He expresses skepticism about the notion that the room is haunted, arguing that a belief in the supernatural reveals a religious and intellectual inferiority (Freeman 1986, p. 133). Yet, when he tries to investigate, he discovers that he cannot even make it through the door: "He might as well have essayed to enter the solid side of a house. He could not believe his senses. The door was certainly open . . . but he could not enter." Finally, in a fit of fear, he runs to his own room "like a terror-stricken girl" (Freeman 1986, p. 157–58). He immediately tells Sophia there is "some accursed evil power at work," which underscores how he, unlike the female characters, evinces certainty about what happened. He does not doubt himself, nor does he wonder whether he will be believed. His inability to enter the room protects him from the haunting and signals to the reader that the anxieties and fears this text expresses is meant for one audience only: women.

The gendered bind that arises from these events—the women's inability to claim their experiences—mirrors the predicament caused by a repressive patriarchal society that leaves no space for women to live authentic lives. The women *know* there is something wrong, but they cannot voice or articulate that sentiment and, as a result, internalize and interpret the discrepancy between what they know to be true and what they can say to be true as inherent and intractable feminine weakness. When they are not struck literally silent—as Amanda Gill, Louisa Stark, Elvira Simmons, and finally Sophia Gill are—they are proven wrong and have to face that they may be losing their sanity.

That the figure of the stereotypical spinster is at the center of what haunts the southwest chamber highlights the importance of gender, space, and economy in this short story. The source of all that ails the house is the spirit of Amanda and Sophia's Aunt Harriet, a dreadful spinster, who in life refused to allow her nieces entrance into their ancestral home because she was angry at her younger sister (their mother) for marrying for love. It is Aunt Harriet's scent that permeates the Southwest chamber, her dresses that appear in the room, and her bonnet that ultimately tries to strangle poor Elvira Simmons. The final haunting that drives Sophia to decide to sell the house is when she looks in the mirror and becomes convinced that instead of her own "middle-aged and good to see" face she sees "the face of a very old woman scowling forever with unceasing hatred at herself and all others, at life, and death, at that which had been and that which was to come" (Freeman 1986, p. 162). Though her sister claims that Sophia looks as she always does, Sophia is so distressed at the mere possibility of becoming Aunt Harriet that she insists that not only will she not sleep in the haunted room again, but that the entire house has to be sold. The price of becoming Aunt Harriet is higher than the price of becoming homeless. The specter of the spinster aunt literally haunts the house, but what she represents is even more important: the potential for Sophia and Amanda to become old and embittered, like their spinster Aunt Harriet. In this way, this story exposes that what haunts single women during this time period is not just economic vulnerability, though that is certainly important. In sharp contrast to the ending of "The New England Nun," where the decision to remain single and alone results in contentment and freedom, for the Gills sister, singlehood means ruin. Marriage is more than an economic agreement,

then; it also represents cultural capital and general well-being, and the figure of the spinster is so horrifying to the sisters that they would rather leave than face it. In contrast to the relatively happy ending of "The Vacant Lot," in which a family moves to a haunted house and ends up selling it and moving back to their beloved original home, Amanda and Sophia Gills, and their niece Flora, are left to fend for themselves, homeless and without any visible means of support.

"The Hall Bedroom," published shortly after "The Southwest Chamber" distinguishes the consequences of singlehood and gender, further emphasizing Freeman's growing awareness of gender discrepancies in the concept of environmental determinism. Moreover, this story reveals that even if domestic space is active, fluid, and even agential, this vital materialism has different consequences for men than it does women. The story begins from the perspective of a "highly respectable woman," a widowed landlady, who has been forced to run a boardinghouse after her husband's death. Much like the Gills sisters, Mrs. Elizabeth Jennings takes boarders not because she wants to but because she must. As in "The Southwest Chamber," the boardinghouse as a whole is successful, except for one troublesome room that ruins everything. The two stories end in the same way: because of the hauntings, Mrs. Jennings is forced to sell her home and move, without any clear sense of where she is going or how she is going to make a living.

The primacy of female characters has dissipated by the time Freeman writes "The Hall Bedroom," however, and the hauntings, if they can even be called that, are remarkably different as well. The bulk of the story is told through the journals of one of the male boarders, Mr. George C. Wheatcroft, a bachelor who finds himself living in one of the most undesirable locations of the house—a hall bedroom on the third floor. At first, Wheatcroft's position seems to mirror that of the turn-of-the-century woman. He is trapped in a small, domestic space that he finds confining and limited. The location and its lack of habitability is one of the first things Wheatcroft notes in his journal, and, as with many of Freeman's characters, his surroundings are consistent with his own identity and selfhood. George's "ignominious and sternly uncompromising" hall bedroom represents the "ignominy of the dweller therein" (Freeman 1992, p. 67). He is "lost in love ... lost in money ... lost in the struggle for preferment ... lost in health and strength" (Freeman 1992, p. 68). Despite this, he views himself as quite apart from women in similar circumstances, as his commentary on his landlady reveals. She is nice enough, but, he writes, "The struggle for money always injures the fine grain of a woman; she is too fine a thing to do it; she does not by nature belong with the gold grubbers, and it therefore lowers her; she steps from heights to claw and scrape and dig. But she cannot help it oftentimes, poor thing, and her deterioration thereby is to be condoned" (Freeman 1992, p. 68).

The distance between single women and single men, both residing in a boardinghouse, becomes even clearer when the uncanniness of the hall bedroom begins to emerge. Mr. Wheatcroft, upon waking up in the middle of the night to take his medicine, discovers that his small hall bedroom has magically expanded in the dark, and as he walks toward his dresser, he finds himself in a new world, where his senses are heightened and the experience of the body is all encompassing. The first night he undergoes this experience, it is the sense of smell that is stimulated—he is immersed in a "ravishing fragrance" that is a rose, and yet not a rose, something more divine than a rose, an experience of "sheer delight, a rapture of sublimated sense" (Freeman 1992, p. 75). Though he does become afraid, it is this experience of complete contentment and sensory pleasure that eventually brings him to believe that he must be in danger. The next nights bring similar joys—of taste, of sound, of sight, and finally, of touch—all of which he details with increasing enthusiasm in his journal. Eventually, Mr. Wheatcroft disappears into the hall bedroom, presumably to enjoy pleasures unavailable to him in real life.

If Mr. Wheatcroft's life is vastly improved by the hall bedroom, it is Mrs. Jennings who must pay the price for that improvement. Indeed, it is Mrs. Jennings who pays the price. She finishes the narrative by explaining that after her tenant disappears, the police come to investigate, even arresting Mr. Wheatcroft's friend, but that nothing comes of the investigation. The owner offers to allow her to stay and continue renting rooms, but she explains that she could not, because even though " ... I was not afraid of anything myself ... my boarders would leave, and I knew I could not get anymore. I

told him I would rather have had a regular ghost than what seemed to be a way of going out of the house to nowhere and never coming back again" ([Freeman 1992](), p. 81). Mrs. Jennings' phrasing, that she does not like how people can leave the house "to nowhere," never to return speaks of something that she is both faced with and that is also inaccessible. The hall bedroom is the ultimate failure of domesticity because it offers the possibility that one can escape the domestic, but to do so is to enter a "nowhere," a space that is too frightening (for her) to even imagine. This story represents, then, an acknowledgment that there may be an escape from the domestic, but it is not an escape available to the spinster. Instead, that potential avenue outward for others results in Mrs. Jennings' expulsion from the home and renders her literally homeless. Mrs. Jennings is not blessed with the option of "never coming back again," insofar as the only place for her to occupy, as a single, older woman in America at the turn of the century is another house where she hopes that her "ill luck" will not follow her. Like "The Southwest Chamber," this story ends with an uninhabitable domesticity.

### 3. "Eleanor Come Home": Shirley Jackson's *The Haunting of Hill House*

On the surface, the hauntings in Freeman's work and the hauntings in Jackson's work seem quite dissimilar. In Freeman's texts, it is the mundane that haunts the female inhabitants. It is more explicit in these texts that it is domesticity and domestic tasks becoming unruly and untamed that troubles the characters. Jackson's supernatural follows tropes more familiar to twenty-first-century readers and viewers familiar with horror movies—loud noises in the middle of the night, cold spots, and possession. Yet beneath those external differences, the core concerns are similar: what do we do when the domestic slips into the undomesticated in a way that makes homes unlivable?

Determining how to interpret Eleanor's deeply rooted desire for home is contingent on how we read houses more broadly, as well as how we read the house in this work, or as a trope within Jackson's oeuvre. Home is at the center of the novel, as well as many of Jackson's other works ([Shotwell 2013](), p. 132). Both of her memoirs, *Life Among the Savages (1948)* and *Raising Demons* (1953) begin with moving to a new house, and the bulk of the narratives occur within the boundaries of the house. While these narratives are ostensibly light hearted, Jackson's frustration and exhaustion peak through the funny anecdotes about refrigerator doors falling off the hinges and the cat that seems perpetually to be having kittens. Betty Friedan critiqued Jackson harshly in *The Feminine Mystique* for the domestic pieces that Jackson published in women's magazines, arguing that "housewife writers" propagated myths of domesticity that kept women locked into toxic domestic roles ([Savoy 2017](), p. 30). To some degree, her memoirs do feel quite distant from her darker novels and short stories; yet, as Eric Savoy points out, Jackson's domestic writings are not that distant from her work that is considered more Gothic, and this closeness belies a darker underside to all of Jackson's work (p. 831). While I would not go so far as to categorize either of her memoirs as Gothic texts, they do reveal a central female narrator who understands all too well why domesticity might need to be divorced from marriage and motherhood if it is meant to foster feminine autonomy.

By the time Jackson wrote *The Haunting of Hill House*, the elements of the Female Gothic took the place of humor in her discussion of domesticity. Eleanor's willingness, eagerness even, to embrace Hill House's "insistent hospitality" is especially interesting, given the cultural milieu of the late 1950s and early 1960s. In many ways, *The Haunting of Hill House* and even *We Have Always Lived in the Castle* can both be interpreted as dire warnings regarding the role of the house and domesticity in women's lives. Just as Freeman was writing within the cultural context of the cult of domesticity, Jackson was writing from within what critics have described as a resurgence of the cult of domesticity ([Hague 2005](), p. 83). As with Freeman, the figure of the house is as deeply desired as much as it is feared. As Hague puts it, houses in Jackson's work are often "places of entrapment and incarceration for the women who visit or live in them" (p. 82).

Much of the critical interpretations of *The Haunting of Hill House* involve the question of the ending: why does Eleanor kill herself, and what does that mean? There are three predominant interpretations: that Eleanor's death is a tragic representation of mental illness and a result of her exile from the group

that has become like family to her; that Eleanor is a victim of the house's malice; or that Eleanor achieves liberation from oppressive social norms through death and madness. The first of these is the most common.[4] If we believe that Eleanor longs for a nuclear family and a sense of belonging above all else, then indeed her choice (if we interpret it as a choice at all) to drive her car into a tree, killing herself instantly, is a tragedy because the reason for that decision is that she has been exiled from her proto-family when Dr. Montague tells her that she must leave. There is certainly evidence for this conclusion, particularly when Eleanor arrives and finds herself belonging to this group of people in a way that she has never felt she belonged before. Furthermore, it is as this sense of belonging unravels that we begin to see Eleanor, too, unraveling. A related reading, and the one most explicitly supported by the plot line of the text, is that Eleanor is driven mad by the house, which is itself evil. This interpretation fits most clearly within the genre of the Gothic, at least in the most simplistic sense. If, however, we believe that what she wants is a house of her own and independence above all else, as Roberts compellingly argues, Eleanor's refusal to leave Hill House can be interpreted as one means by which she can get what she desires, for by killing herself, she is potentially united with the house and allowed to stay there indefinitely (Roberts 2017, p. 73). This interpretation, too, has merit, particularly when we look at the beginning of the text and Eleanor's fantasies about occupying cottages with white curtains and stone lions. Yet none of these interpretations are entirely satisfying because it seems to be a case of both/and, rather than either/or. I interpret the significance of the house in this narrative as both a representation of Eleanor's psychological longing to create a sense of belonging *and* also a reflection of a desire (akin to that expressed by Freeman's work) to have an idealized domestic space that simultaneously enables and preserves feminine identity and autonomy *and* the concurrent anxiety that such a space is impossible within patriarchy. The opening sequence of *The Haunting of Hill House* articulates the stakes of the novel: Eleanor finding a place that she can claim as home.[5] Though the house is described, initially, as evil, as a house that is "chillingly wrong" (Jackson 1984, p. 40), it is also not entirely dissimilar from the earlier fantasy houses. Like the earlier houses Eleanor imagines, Hill House is detached from the rest of the world, set back and seemingly different from all that surrounds it (p. 49). This isolation is important, as Roberts (2017) points out, explaining that: "Taken together with Eleanor's longing for a home, a safe space, and isolation, Jackson's novel offers a discourse in which the family is rotten, but the home is safe, and in which enclosure, seclusion, and isolation surface as strategies for happiness and safety" (p. 77). Indeed, Hill House "steadies and locates" the group assembled inside (Jackson 1984, p. 58), and they are encouraged to "arrange things to please" themselves, just as Eleanor imagines her own fictitious apartment, where everything will be exactly the way she wants it to be (Jackson 1984, p. 63). From the first night, Eleanor senses that the house will not "let [them] go" (Jackson 1984, p. 75) and rather than finding that frightening, she seems to take comfort in that sense of inevitability. She thinks, "I am here . . . and I have a place in this room. I have red shoes and tomorrow I will still be here" (Jackson 1984, p. 83). The key phrase is that she "has a place" in Hill House, an experience that is lacking in the rest of her life. She is, for the first time, "emplaced," in Casey's words. Later, she repeats this act of emplacement, telling herself, "I am here, I am here" (Jackson 1984, p. 141). It is the sense of "here-ness" that emplaces Eleanor in Hill House, and the fear of being cast out, or displaced, seems to be more powerful than the fear of losing her autonomy and ability to *choose* whether she wants to stay or not. She imagines that the trees and the wildflowers see her as "a creation so unfortunate as not to be rooted in the ground, forced to go from one place to another, heartbreakingly mobile" (Jackson 1984, p. 180). Being emplaced, ensconced in a

---

[4]  See, for example, Eric Savoy; Judie Newman's "Shirley Jackson and the Reproduction of Mothering: *The Haunting of Hill House*;" Roberta Rubenstein's "House Mothers and Haunted Daughters: Shirley Jackson and the Female Gothic;" Michael Wilson's "Shirley Jackson's *The Haunting of Hill House*;" John G. Parks' "Chambers of Yearning: Shirley Jackson's Use of the Gothic;" and Andrew Smith's "Children of the Night: Shirley Jackson and the Female Gothic."

[5]  The trope of domesticity and houses is consistent throughout Jackson's work, but especially in *The Haunting of Hill House* and *We Have Always Lived in the Castle*. See Roberts' for a thoughtful analysis of how Jackson's domestic memoirs merge with her Gothic novels and stories.

home, is better than being "heartbreakingly mobile." In this way, Eleanor's experience echoes that of the characters in Freeman's stories because even though they are all living in the non-home of a boarding house, this is nonetheless preferable to not having a home at all.

It is clear that while others may find Hill House distressing, Eleanor finds it just the opposite. She has her first good night's sleep in years her first night there, and she notes to herself that she finds the house "charming" (Jackson 1984, p. 95). The second morning she wakes up to find herself "unbelievably happy" (Jackson 1984, p. 136). Even after the house increases its antagonism, Eleanor notes that she feels an "overwhelming wild happiness" (Jackson 1984, p. 180). It is not just the house, though. Part of Eleanor's pleasure derives from the fact that she feels, for the first time in her life, that she belongs to a community of people, calling them her "family" (Jackson 1984, p. 97). This pleasure, though, is tinged with her constant fear that she will be left out or left behind, that she will be exiled from the group. When Theodora comments that she thinks it might be best if Eleanor goes "home," Eleanor immediately resists, saying that she does not want "to go" (Jackson 1984, p. 117). Significantly, Eleanor does not say that she does not want to "go home" but rather that she does not want "to go." She is already "home" in Hill House, and this sense of home-ness is emphasized by the house's continued messages (perhaps written in blood) saying "HELP ELEANOR COME HOME ELEANOR" (Jackson 1984, p. 155). "Home" is Eleanor's core desire, which becomes even clearer when Mrs. Montague reads her transcript from her work with the planchette. The mysterious speaker identifies herself as "Nell . . . Eleanor Nellie Nell Nell" and when she is asked what she wants, she replies "Home." To the question, "Do you want to go home?", Planchette replies, "Want to be home" (Jackson 1984, p. 192), emphasizing that Eleanor does not need to *go* anywhere to be home in Hill House.

Eleanor, however, is not necessarily attached to the idea of staying at Hill House after everyone else leaves. Eleanor proposes that when she leaves Hill House, she will follow Theodora home and live with her because she wants "to be someplace where I belong." When Theodora rejects this plan and asks Eleanor if she "always" goes "where [she's] not wanted," Eleanor replies: "I've never been wanted *anywhere*" (Jackson 1984, p. 209). We see, in this moment, Eleanor's profound vulnerability. Her desire to feel like she belongs somewhere is so overwhelming that she is willing to insert herself into another person's life, even knowing that she is not wanted. We know that, up to this point, she has never had a home of her own: for the first part of her adulthood, she cared for her ailing mother (whose death continues to haunt her in the form of the insistent knocking at Hill House) and later lived with her sister and brother-in-law. Thus, what is at stake for Eleanor is not an idealized notion of middle-class domesticity. She is not longing for social acceptance within a broader scope, or a perfectly appointed house. Nor is she attempting to gain a traditional marriage and family. As critic Shotwell (2013) points out, her proposal to come live with Theodora is an expression of her "unspoken and unspeakable love for Theo." Shotwell reads Theo's rejection of Eleanor as the prelude to Eleanor's increasingly "queered desire for the house itself" (p. 136). I interpret this moment similarly but would add that it is not simply that Eleanor's desire transfers from one person to the next and then, when that fails, to the house. Instead, I read this shift as revealing that what Eleanor wants, more than anything, is a sense of emplaced belonging. Her longing for home is simultaneously emotional, psychological, and embodied—she wants to feel a sense of belonging, but that belonging has to be, for Eleanor, emplaced. The human connection within that place is secondary, not primary. In short, I interpret Eleanor's desire to go home with Theo as being another expression of her longing for a home, not necessarily a desire for Theo, or any one person.

The most straightforward reading, perhaps, of this novel and this particular haunted house is that Eleanor's sense of rightness in the house and sense of belonging is representative of exactly what's *wrong*. Dr. Montague, at the very beginning, describes the house as having "insistent hospitality" (Jackson 1984, p. 67), and certainly midway through the novel, Eleanor seems to be sucked into that devouring, all-consuming "hospitality." When she describes the knocking at their door, she explains, "The sense was that it wanted to consume us, take us into itself, make up a part of the house, maybe—oh, dear. I thought I knew what I was saying, but I'm doing it very badly" (p. 139). The house echoes

Eleanor's maternal trauma, the knocking in the house is akin to Eleanor's mother knocking on the wall the night that Eleanor did not wake up to give her mother her medicine. Certainly, one of the issues that haunts Eleanor is her sense that caring for her overbearing mother has consumed her life, in the same way that Hill House now threatens to consume her. Rather than resist this, however, Eleanor embraces it, desiring to be taken fully into the house, if only so that she can belong somewhere. Wilson (2015) describes the house's behavior toward Eleanor as a "seduction," explaining that the real tragedy of the novel is that Eleanor "fall[s] out of the dream into a glimpse of the unveiled reality of the universe-as-House in her last moments" (p. 121). In the same vein, Anderson (2009) reads the house as a "symbol of patriarchal domination" that "possess[es] and then kill[s]" Eleanor (p. 200). The idea that the house is a malevolent force that, in particular, harms the female inhabitants is borne out by the history of the house, in which nearly all the women who live there are driven to madness and/or suicide (Anderson 2009, p. 201).

When reading Jackson alongside Freeman, especially given both of their interests in domesticity as a site for feminine autonomy and freedom from societal, heterosexist restraints, it is tempting to read Eleanor's longing for home as a feminist attempt at independence and agency. Ultimately, though, the text resists this reading, albeit unintentionally because the price of "being home" is annihilating the self and any sense of agency. When Eleanor tries to explain how the initial "HELP ELEANOR COME HOME ELEANOR" message from the house makes her feel, she says that she knows she will not get hurt, if she can only "surrender" (Jackson 1984, p. 160). This trope emerges again during one of the house's final outbursts, when Eleanor realizes that she must "surrender" herself entirely: "No; it is over for me. It is too much, she thought, I will relinquish my possession of this self of mine, abdicate, give over willingly what I never wanted at all; whatever it wants of me it can have" (p. 204). The home Hill House creates is not a space for Eleanor to flourish; it is a space that demands that she give over all of herself, a self that she identifies is something she "never wanted at all." Moreover, it is a space that is identified quite clearly with the maternal, and more specifically Eleanor's own overbearing mother. As Pascal (2014) puts it, "Unquestionably for Eleanor the allure of the house, and also its horror, is bound up with the sense that it wishes to envelop her in a maternal embrace so comprehensive that her newly won independence and all vestiges of her individuality will be subsumed utterly" (p. 469). Eleanor's desire to be subsumed, by the house and/or the mother, is exactly what draws her to the house.

What is most troubling about Hill House is simultaneously what is most pleasing about it. The house removes autonomy for its inhabitants, but it does so in a way that promises a constant sense of home and belonging. The house and its inhabitants become so enmeshed that one slips into the other. Within this novel, it becomes impossible to disentangle the agency of the house from the agency of the people living in the house; they converge and blend. Eleanor invites that merging because she (mis)interprets it as a way to find a place that she will always belong. Jackson's representation of this kind of domesticity takes on the quality of a nightmare, because ultimately, Eleanor hopes that ceding her own agency will provide her with protection from an unruly and uninhabitable world.

## 4. Conclusions: The Tyranny of the Home

The embedded desire in both Mary Wilkins Freeman and Shirley Jackson's texts is that domestic space will function as promised: as a separate sphere that is protected from the outside world, both economic and natural, where women can live and flourish and create a sense of coherent feminine subjectivity. They engage with the traditions of the Female Gothic throughout their works, while also departing from them by removing the conventions of marriage and motherhood from domesticity. It is almost as if they imagine that the ideologies of environmental determinism and separate spheres could affirm and liberate women, if only the literal patriarch were removed from the scene. Ultimately, though, even if the specter of marriage and motherhood can be excised from domesticity and domestic spaces, the specter of patriarchy cannot be done away with so easily. For Freeman's characters, independent domesticity is dangerous because of its precarity; if domesticity cannot be performed properly, even if it is through no fault of her own, homelessness is always a danger for a woman, if not

a reality. Her female characters in "The Southwest Chamber" and "The Hall Bedroom" do not have the economic means, or the cultural support, to maintain independent domesticity. While they may be protected from "male anger" in Ellis's (1989) words (p. xi) that haunts many women in the genre of the Female Gothic, they are not immune from the pressures of the marketplace and the cultural forces that limit women's abilities to garner economic independence. For Jackson's character, Eleanor, the price of having an emplaced sense of identity as an independent woman is being subsumed entirely, to the point of insanity and/or death, and even though not a husband, Dr. Montague looms large in the background of the novel as a father figure who ultimately expels her from the space she had hoped to make her home. Their texts are haunted by the realization that the ideology of separate spheres is an unattainable myth and that, moreover, the price of having such a domestic space is too high to pay. Home and emplacement, for these characters, and perhaps authors, cannot be disentangled from the patriarchal ideologies of domesticity, and thus their fantasies of emplacement will always remain just that—fantasies.

**Funding:** This research received no external funding.

**Conflicts of Interest:** The author declares no conflict of interest.

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
