# Peer review of "The Domestic Tyranny of Haunted Houses in Mary Wilkins Freeman and Shirley Jackson"

_humanities, doi:10.3390/h8020107_

Reviewer 1 Report

Humanities  - the domestic  tyranny of haunted houses

This article uses unruly domesticity to explore the haunted house narratives of Freeman and Jackson who look at domestic space for single women and it considers, in particular, The Haunting of Hill House and ‘A New England Nun’.   The argument is that the emplacement within a domestic context is essential to securing female subjectivity and the authors then intend to link this to the haunted house narratives and  this is largely achieved.

 ‘provide a refuge’ p2, is one of the slightly ungainly expressions, as is ‘haunted by the (25) anxiety that while embracing “blessed singlehood” housed securely within blissful domesticity is a (26) tantalizing prospect, this ideal can be frighteningly inaccessible.’p1 .  So I suggest a further close read through the work to ensure the expression is clear. 

The discussion of the ways in which domestic objects behave in an unruly defamiliarised way showing they cannot be trusted needs to be more explicitly linked to Freeman’s questioning of the right? Ability? Safety of women (spinsters) attempting to establish both economic stability and a domestic security.  I remain unsure of the argument although the close reading is good and the piece sound, the argument about  the haunted domestic spaces (poltergeist behaviour) makes me  question -might the forces attacking the two women in Freeman’s story be trying to undermine their freedom and ability (as is mentioned earlier) to stay solvent (rather than making big profits) so it is preventing them from surviving, punishing them for going against the norm of domesticity being only for married women.  Is this reading it a little more radically than the author of the piece, as preventing economic independence? And if so is Freeman showing its vulnerability and instability? But is Freeman also reinforcing the constraints and restrictions on women’s economic freedoms and replaying the hysterical endings they face in fictions contemporary to that period if the ‘figure of the stereotypical spinster is what haunts is this fire? attacking and undermining their attempt to be domestic or their attempts to combine that with paying the bills and   in going against society’s expectation of all women to be dependent on men and married? 

Perhaps a little further clarity in the fundamental arguments about the role of the haunting, and the authors’ arguments in that context – time and place -would make this a stronger piece.  The arguments about Jackson writing at a time of the resurgence of the cult of domesticity and ostensibly from within but actually underpinning it are more crafted.  Jackson’’s work as a whole regularly problematizes the safety of the domestic.  The statement ‘For both of these authors, in other words, a blissful and (363) independent domesticity is deeply desired, but their haunted house works demonstrate an anxiety (364) that the promises of domesticity are false and that escaping heterosexual unions doesn’t mean that (365) one can escape the oppressiveness of patriarchy.’ is very clear and might open the article in some form to guide the reader more overtly. 

Eleanor seems to be a romantic with her journeys ending with lovers phrases and this is at odds with the need for belonging in a community and torn in several ways between the expectations of the time for women, romance, domesticity, desire for new freedoms, financial independence and so on, with the spectre of mental illness reinforced by her social insecurity.

The argument about the houses’ insistent hospitality is very interesting – is it entrapping and devouring? 

How does the haunting work in Jackson’s novel?

I really enjoyed reading this and the arguments about the desire for a domesticity which is also removed from the economic stringencies of spinsterhood and of the larger paternalistic society is a convincing one.  I should like to see more about the actual haunting and the disorder of direction, of ostensible safe places, and the ways in which Eleanor’s socially constructed fantasies about romance and belonging are undermined by her guilt about her mother – one version of home perhaps undermining the other when she actually has no home at all.  It  could  be  enhanced also  by a closer discussion of the Gothic elements of the two pieces in relation to the ways the tales deal with women’s attempts at independence under patriarchy (financial and psychological) and the challenge that economic and domestic independence offers to that established set of behaviours and beliefs so that perhaps there is a war in the women themselves which presents itself in the disruption of their external contexts – rooms, clothes etc.

Stepping back and fully clarifying the argument in its economic historical context and theorising the gothic elements further would make this a stronger piece.

Author Response

Point 1: 'provide a refuge' p2, is one of the slightly ungainly expressions, as is 'haunted by the (25) anxiety that while embracing "blessed singlehood" housed securely within blissful domesticity is a (26) tantalizing prospect, this ideal can be frighteningly inaccessible.’p1 .  So I suggest a further close read through the work to ensure the expression is clear. 

Response 1: I’ve revised those specific phrases, as well as done a close reading of the rest of the essay in order to eliminate wordy sentences and unclear phrasing. 

Point 2: The discussion of the ways in which domestic objects behave in an unruly defamiliarised way showing they cannot be trusted needs to be more explicitly linked to Freeman’s questioning of the right? Ability? Safety of women (spinsters) attempting to establish both economic stability and a domestic security.  I remain unsure of the argument although the close reading is good and the piece sound, the argument about  the haunted domestic spaces (poltergeist behaviour) makes me  question -might the forces attacking the two women in Freeman’s story be trying to undermine their freedom and ability (as is mentioned earlier) to stay solvent (rather than making big profits) so it is preventing them from surviving, punishing them for going against the norm of domesticity being only for married women.  Is this reading it a little more radically than the author of the piece, as preventing economic independence? And if so is Freeman showing its vulnerability and instability? But is Freeman also reinforcing the constraints and restrictions on women’s economic freedoms and replaying the hysterical endings they face in fictions contemporary to that period if the ‘figure of the stereotypical spinster is what haunts is this fire? attacking and undermining their attempt to be domestic or their attempts to combine that with paying the bills and   in going against society’s expectation of all women to be dependent on men and married? 

Response 2:I’m intrigued by this line of questioning – I think that in the case of this story, it is suggested that they are being punished for their failure to adhere to cultural norms, but it almost seems as if what really frightens them is the possibility of becoming a lifelong spinster, which is represented as a figure of horror. In other words, I think the characters have internalized the cultural norms to a degree that is not present in Freeman’s earlier work. I articulated this in two sections of my analysis of the story: 249-251 and 281-289. 

Point 3: Perhaps a little further clarity in the fundamental arguments about the role of the haunting, and the authors’ arguments in that context – time and place -would make this a stronger piece.  The arguments about Jackson writing at a time of the resurgence of the cult of domesticity and ostensibly from within but actually underpinning it are more crafted.  Jackson’’s work as a whole regularly problematizes the safety of the domestic.  The statement ‘For both of these authors, in other words, a blissful and (363) independent domesticity is deeply desired, but their haunted house works demonstrate an anxiety (364) that the promises of domesticity are false and that escaping heterosexual unions doesn’t mean that (365) one can escape the oppressiveness of patriarchy.’ is very clear and might open the article in some form to guide the reader more overtly. 

Response 3: 

I’ve revised the end of the introduction to reflection this suggestion. It now reads: Freeman and Jackson depart from this tradition by focusing their narratives on what haunts the spaces of spinsters, women who have potentially escaped some of the most restrictive aspects of patriarchy. These two women writers recognize the limitations of environmental determinism and the separate spheres doctrine, but they also try to harness tendrils of subversive potential within these concepts.[i]Both authors assert that emplacement within domestic enclosure is essential to securing feminine subjectivity, but their haunted house narratives undermine that very emplacement. Freeman’s stories chart out the material barriers to home for single women, while Jackson’s novel suggests that the barriers to independent identity and domesticity are primarily interior and psychological. For both of these authors, in other words, a blissful and independent domesticity is deeply desired, but their haunted house works demonstrate an anxiety that the promises of domesticity are false and that escaping heterosexual unions doesn’t mean that one can escape the oppressiveness of patriarchy. Reading their haunted house narratives together reveals the dynamics and tensions of this domesticity and the feminist potential such sites engender, even if the characters in question cannot fully realize that potential.

Point 4: Eleanor seems to be a romantic with her journeys ending with lovers phrases and this is at odds with the need for belonging in a community and torn in several ways between the expectations of the time for women, romance, domesticity, desire for new freedoms, financial independence and so on, with the spectre of mental illness reinforced by her social insecurity.

The argument about the houses’ insistent hospitality is very interesting – is it entrapping and devouring? 

How does the haunting work in Jackson’s novel?

Response 4: The house is described as consuming. I emphasized this more in line 496 and addressed issues of maternal haunting in lines 499-505.

Point 5:   I should like to see more about the actual haunting and the disorder of direction, of ostensible safe places, and the ways in which Eleanor’s socially constructed fantasies about romance and belonging are undermined by her guilt about her mother – one version of home perhaps undermining the other when she actually has no home at all.  It  could  be  enhanced also  by a closer discussion of the Gothic elements of the two pieces in relation to the ways the tales deal with women’s attempts at independence under patriarchy (financial and psychological) and the challenge that economic and domestic independence offers to that established set of behaviours and beliefs so that perhaps there is a war in the women themselves which presents itself in the disruption of their external contexts – rooms, clothes etc.

Response 5: I address issues of maternal haunting in lines 499-505, and the revisions of the introduction and conclusion requested by Reviewer 2 more firmly place this essay within the context of scholarship of the Female Gothic. 

Point 6: Stepping back and fully clarifying the argument in its economic historical context and theorising the gothic elements further would make this a stronger piece. 

Response 6: See above notes about the revision of the introduction and conclusion to situate my argument in clearer conversation with the existing scholarship. 

Reviewer 2 Report

This article is an important contribution to current scholarly conversations about women writing the supernatural. Your argument provides a perspective that complicates debates about the feminist potential of the Gothic genre. The conception of haunted houses as anti-domestic spaces also recalls Freud's notion of the uncanny as Das Unheimlich (unhomely, unfamiliar). This piece also works to help establish a tradition of women writers using supernatural genres for subversive critique that spans literary and historical periods.

Proofread for occasional errors (e.g. unfinished sentence at line 69)

Author Response

Point 1: Proofread for occasional errors (e.g. unfinished sentence at line 69).

Response 1: This sentence has been corrected. In addition, I have proofread the manuscript to eliminate wordy phrasing and other minor errors. 

Reviewer 3 Report

This is a well-argued and insightful paper on two key writers of American Gothic fiction. While the topics of domesticity and women's oppression in these texts have been covered extensively already within Gothic and feminist literary scholarship, this paper does make a valuable contribution to the ongoing debate about the affirmative/subversive cultural work of Gothic fiction.

In terms of revision, I would like to see the author include some engagement with Shirley Jackson's memoirs, Life Among the Savages and Raising Demons, which are very enlightening when  it comes to the topic of domesticity and womanhood. Some explicit discussion of the long-stranding tradition of the Gothic genre's engagement with domestic ideology would be useful in the introduction, if only to highlight what the thesis of this paper adds to the scholarship extant. I am thinking of Ferguson Ellis's classic study The Contested Castle, or Bailey's American Nightmares: The Haunted House Formula in American Popular Fiction.

It remains somewhat unclear to me what the concept "domestic ecology" (line 213) means in the context of this paper. the term "ecology" seems borrowed from a different register and does little to really clarify the critical point being made.

The section on "The Hall Bedroom" could be streamlined a little as there some unnecessary plot summary in these paragraphs in my view.

I think the concluding paragraph can be expanded to include some reflection on the significance of gothic genre conventions to the exploration of the theme of domesticity.

Author Response

Point 1:  In terms of revision, I would like to see the author include some engagement with Shirley Jackson's memoirs, Life Among the Savages and Raising Demons, which are very enlightening when  it comes to the topic of domesticity and womanhood

Response 1: I added the following paragraph (383-399) to address this concern: Determining how to interpret Eleanor’s deeply rooted desire for home is contingent on how we read houses more broadly, as well as how we read the house in this work, or as a trope within Jackson’s oeuvre. Home is at the center of the novel, as well as many of Jackson’s other works (Shotwell 2013, p. 132). Both of her memoirs, Life Among the Savages (1948) and Raising Demons(1953) begin with moving house, and the bulk of the narratives occur within the boundaries of the house. While these narratives are ostensibly light-hearted, Jackson’s frustration and exhaustion peak through the funny anecdotes about refrigerator doors falling off the hinges and the cat that seems perpetually to be having kittens. Betty Friedan critiqued Jackson harshly in The Feminine Mystiquefor the domestic pieces that Jackson published in women’s magazines, arguing that “housewife writers” propagated myths of domesticity that kept women locked into toxic domestic roles (Savoy 2017, p. 30). To some degree, her memoirs do feel quite distant from her darker novels and short stories; yet, as Eric Savoy points out, Jackson’s domestic writings are not that distant from her work that is considered more Gothic, and this closeness belies a darker underside to all of Jackson’s work (p. 831). While I would not go so far as to categorize either of her memoirs as Gothic texts, they do reveal a central female narrator who understands all too well why domesticity might need to be divorced from marriage and motherhood if it is meant to foster feminine autonomy. 

                  By the time Jackson wrote The Haunting of Hill House, the elements of the Female Gothic took the place of humor in her discussion of domesticity.

Point 2: Some explicit discussion of the long-stranding tradition of the Gothic genre's engagement with domestic ideology would be useful in the introduction, if only to highlight what the thesis of this paper adds to the scholarship extant. I am thinking of Ferguson Ellis's classic study The Contested Castle, or Bailey's American Nightmares: The Haunted House Formula in American Popular Fiction.

Response 2:I incorporated a paragraph at the end of the introduction, lines 135-159 to address these suggestions. The two final paragraphs now read: The doctrine of separate spheres and its implications for women plays a prominent role within the tradition of the Female Gothic. Kate Ferguson Ellis identifies the eighteenth century Gothic novel as a site of critique of the doctrine of separate spheres because these works illuminate the tension between the home as a “refuge from evil” and the home as a “prison” (1989, p. xiii, p. xv). Carpenter and Kolmar (1991) argue that the figure of the haunted house permeates the writing of women writers in this genre because the doctrine of separate spheres so closely identified women with domestic spaces, and women writing in the supernatural genre “inherited a series of themes and images - of women victimized in their own homes, of women dispossessed of home and property, of the necessity of understanding female history, and of the bonds between women, living and dead, which help to ensure women’s survival” (p. 14, 10). Likewise, Dale Bailey (1999) calls the (haunted) house a “potent symbol” that’s a “primary marker of class and our central symbol of domesticity” (p. 8). While Jeffrey Weinstock (2008) acknowledges the importance of uncanny domestic space in American women’s supernatural writing in the nineteenth century, he also asserts that two of the primary themes that haunt this writing are “marriage and motherhood” (p. 22). Indeed, within the Female Gothic, haunted domestic space is often inextricable from an accompanying marriage and the duties of mothering, and critiques of domesticity are partially, at least, critiques of the way marriage and motherhood within a patriarchal society place profound limitations on women. 

                  Freeman and Jackson depart from this tradition by focusing their narratives on what haunts the spaces of spinsters, women who have potentially escaped some of the most restrictive aspects of patriarchy. These two women writers recognize the limitations of environmental determinism and the separate spheres doctrine, but they also try to harness tendrils of subversive potential within these concepts.[i]Both authors assert that emplacement within domestic enclosure is essential to securing feminine subjectivity, but their haunted house narratives undermine that very emplacement. Freeman’s stories chart out the material barriers to home for single women, while Jackson’s novel suggests that the barriers to independent identity and domesticity are primarily interior and psychological. For both of these authors, in other words, a blissful and independent domesticity is deeply desired, but their haunted house works demonstrate an anxiety that the promises of domesticity are false and that escaping heterosexual unions doesn’t mean that one can escape the oppressiveness of patriarchy. Reading their haunted house narratives together reveals the dynamics and tensions of this domesticity and the feminist potential such sites engender, even if the characters in question cannot fully realize that potential.    

The appropriate citations have also been added to the References page.

Point 3: It remains somewhat unclear to me what the concept "domestic ecology" (line 213) means in the context of this paper. the term "ecology" seems borrowed from a different register and does little to really clarify the critical point being made.

Response 3:I have removed the term “domestic ecology” and replaced it with “domesticity” in lines 14 and 206.

Point 4: The section on "The Hall Bedroom" could be streamlined a little as there some unnecessary plot summary in these paragraphs in my view.

Response 4: I deleted the paragraph that originally appeared between lines 312-313 and replaced it with a sentence that summarizes the ending of the story. 

Point 5:   I think the concluding paragraph can be expanded to include some reflection on the significance of gothic genre conventions to the exploration of the theme of domesticity.

Response 5: I expanded the final paragraph, so that it now reads:
The embedded desire in both Mary Wilkins Freeman and Shirley Jackson’s texts is that domestic space will function as promised: as a separate sphere that is protected from the outside world, both economic and natural, where women can live and flourish and create a sense of coherent feminine subjectivity. They engage with the traditions of the Female Gothic throughout their works, while also departing from them by removing the conventions of marriage and motherhood from domesticity. It is almost as if they imagine that the ideologies of environmental determinism and separate spheres could affirm and liberate women, if only the literal patriarch were removed from the scene. Ultimately, though, even if the specter of marriage and motherhood can be excised from domesticity and domestic spaces, the specter of patriarchy cannot be done away with so easily. For Freeman’s characters, independent domesticity is dangerous because of its precarity; if domesticity cannot be performed properly, even if it is through no fault of her own, homelessness is always a danger for a woman, if not a reality. Her female characters in “The Southwest Chamber” and “The Hall Bedroom” do not have the economic means, or the cultural support, to maintain independent domesticity. While they may be protected from “male anger” in Ellis’s (1989) words (p. xi) that haunts many women in the genre of the Female Gothic, they are not immune from the pressures of the marketplace and the cultural forces that limit women’s abilities to garner economic independence. For Jackson’s character, Eleanor, the price of having an emplaced sense of identity as an independent woman is being subsumed entirely, to the point of insanity and/or death, and even though not a husband, Dr. Montague looms large in the background of the novel as a father-figure who ultimately expels her from the space she had hoped to make her home. Their texts are haunted by the realization that the ideology of separate spheres is an unattainable myth and that, moreover, the price of having such a domestic space is too high to pay. Home and emplacement, for these characters, and perhaps authors, cannot be disentangled from the patriarchal ideologies of domesticity, and thus their fantasies of emplacement will always remain just that—fantasies. 

Round  2

Reviewer 1 Report

The  author(s)  have  taken  revision suggestions into account in their rewrite of what  was   an intriguing  and good paper. The  clarifying of the  issue about  seeking   individual  space  and  a version of domesticity while being haunted  by a fear of spinsterhood   greatly  strengthens  the originality of the piece. It is  very interesting and will make a good contribution to  the field.

Reviewer 3 Report

no further comment or suggestions.